# Stress Can Induce Bovine Alpha-Herpesvirus 1 (BoHV-1) Reactivation from Latency

**DOI:** 10.3390/v16111675

**Published:** 2024-10-27

**Authors:** Fouad El-Mayet, Clinton Jones

**Affiliations:** 1Department of Veterinary Pathobiology, College of Veterinary Medicine, Oklahoma State University, Stillwater, OK 74078, USA; fouad.elmayet@okstate.edu; 2Department of Virology, Faculty of Veterinary Medicine, Benha University, Benha 74078, Egypt

**Keywords:** Bovine alpha-herpesvirus 1 (BoHV-1), reactivation from latency, glucocorticoid receptor (GR), Krüppel-like factor (KLF), specificity protein 1 (Sp1), Wnt/β-catenin/Akt signaling axis

## Abstract

Bovine alpha-herpesvirus 1 (BoHV-1) is a significant problem for the cattle industry, in part because the virus establishes latency, and stressful stimuli increase the incidence of reactivation from latency. Sensory neurons in trigeminal ganglia and unknown cells in pharyngeal tonsils are important sites for latency. Reactivation from latency can lead to reproductive problems in pregnant cows, virus transmission to young calves, suppression of immune responses, and bacterial pneumonia. BoHV-1 is also a significant cofactor in bovine respiratory disease (BRD). Stress, as mimicked by the synthetic corticosteroid dexamethasone, reproducibly initiates reactivation from latency. Stress-mediated activation of the glucocorticoid receptor (GR) stimulates viral replication and transactivation of viral promoters that drive the expression of infected cell protein 0 (bICP0) and bICP4. Notably, GR and Krüppel-like factor 15 (KLF15) form a feed-forward transcription loop that cooperatively transactivates immediate early transcription unit 1 (IEtu1 promoter). Two pioneer transcription factors, GR and KLF4, cooperatively transactivate the bICP0 early promoter. Pioneer transcription factors bind silent viral heterochromatin, remodel chromatin, and activate gene expression. Thus, we predict that these novel transcription factors mediate early stages of BoHV-1 reactivation from latency.

Recent reviews by our lab discussed the role BoHV-1 plays during BRD, how infection influences host immune responses [1], and the role stress plays during reactivation [2]. Although these topics are summarized in this review, the focus of this review is to discuss differences between viral gene expression during early stages of reactivation in the pharyngeal tonsil and trigeminal ganglia. Furthermore, we provided a new discussion summarizing how cell cycle regulators (E2F1, E2F2, and E2F3) and specificity protein 1 (Sp1) cooperate with GR to cooperatively transactivate the immediate early transcription unit 1 promoter. This promoter drives the expression of the two most important viral transcriptional regulatory proteins.

## 1. Pathogenesis of BoHV-1

Bovine respiratory disease (BRD) is initiated by stress and/or virus infection, including by bovine alpha-herpesvirus 1 (BoHV-1), bovine respiratory syncytial virus (bRSV), bovine parainfluenza type 3 virus (PI3V), bovine coronavirus (BoCV), influenza D virus (IDV), and bovine viral diarrhea virus (BVDV), which are reviewed in [3].

Approximately 75% of morbidities and 50% of mortalities in feedlot cattle are linked to BRD [4,5,6,7]. The most common bacterial strains associated with BRD are Mannheimia haemolytica, Pasteurella multocida, Histophilus somni, and Mycoplasma bovis [8]. Mannheimia haemolytica is a commensal bacterium [9] that is present in the normal flora in the upper respiratory tract of healthy ruminants [10]. Hence, BRD is a polymicrobial disease that continues to be the most economically important disease affecting beef and dairy cattle. Stress that can lead to immune suppression and/or co-infections disrupt this commensal relationship [11]. Consequently, Mannheimia haemolytica is frequently present in bronchopneumonia in BRD cases [12,13,14,15].

BoHV-1 acute infection generally causes upper respiratory tract disease [16,17], erodes mucosal surfaces of the upper respiratory tract, enhances colonization of Mannheimia haemolytica in the lower respiratory tract [12,13,14], and promotes interactions between the Mannheimia haemolytica leukotoxin and bovine peripheral blood mononuclear cells, including neutrophils [18,19]. Co-infection of calves with BoHV-1 and Mannheimia haemolytica consistently leads to life-threatening pneumonia [20].

In addition to BRD, BoHV-1 is the most frequently diagnosed cause of viral abortion in cattle located in North America [21,22,23,24,25]. These studies also concluded that BoHV-1 modified live vaccines can induce abortions. While acute infection is responsible for certain abortions, it is likely that reactivation from latency plays a major role in abortions and other reproductive issues.

## 2. BoHV-1 Productive Infection

High levels of virus production occur during acute BoHV-1 infection, which induces apoptosis and inflammation, as reviewed in [3]. Viral gene expression occurs in three well-defined phases: immediate early (IE), early (E), and then late (L). IE transcription unit 1 (IEtu1) encodes two transcriptional regulatory proteins (bICP0 and bICP4), which stimulate productive infection [26,27,28]. IEtu2 encodes the bICP22 protein [27]. A tegument protein, VP16, interacts with two cellular proteins, Oct-1 and host cell factor 1 (HCF-1), and this complex interacts with and specifically transactivates all IE promoters [29,30]. E proteins are generally non-structural and are involved with viral DNA replication. L proteins comprise the infectious virus.

The genomes of *Alphaherpesvirinae* subfamily members, BoHV-1 and Herpes Simplex Virus 1, for example, generally have a high GC content. Many viral promoters within these genomes include Sp1 consensus binding sites and additional GC-rich motifs [31,32]. Sp1 is a zinc-finger transcription factor expressed in all cells, and is an important basal transcription factor for many promoters [33]. Sp1 is bound to Herpes Simplex Virus 1 IE, E, and early/late (E/L) promoters throughout the viral genome, and there are higher levels of binding prior to the onset of viral DNA replication [32,34]. Furthermore, infection of permissive cells with BoHV-1 or Herpes Simplex Virus 1 increases Sp1 and Sp3 steady-state protein levels, and, as expected, these proteins were primarily detected in nuclear extracts of infected cells [35,36]. Silencing Sp1 expression with a specific siRNA or with Mithramycin A, an antibiotic with anti-tumor activity that preferentially interacts with GC-rich DNA, significantly reduced Herpes Simplex Virus 1 replication, indicating that it has antiviral activity [35,36].

## 3. BoHV-1 Inhibits Immune Responses During Productive Infection

BoHV-1 impairs immune responses during acute infection because it suppresses cell-mediated immunity [37,38,39], CD8+ T-cell recognition of infected cells [40,41,42,43], and CD4+ T-cell functions. CD4+ T-cell function is impaired during acute infection of calves because BoHV-1 infects CD4+ T cells and induces apoptosis [44,45]. Fluorescence-activated cell sorting analyses revealed that CD4+ and CD8+ T cells decreased in lymph nodes and PBMCs after infection. The decrease in CD4+ T cells correlates with increased apoptosis.

Infected cell protein 0 (bICP0) inhibits interferon signaling by more than one mechanism [46,47]. For example, the C_3_HC_4_ Zinc RING finger domain of bICP0 is crucial for inducing IRF3 degradation, suggesting that bICP0 ubiquitinates interferon response factor 3 (IRF3). A functional proteasome is important for bICP0-induced IRF3 degradation, which supports the role of the bICP0 RING finger in mediating IRF3 degradation [47]. The capacity of bICP0 to inhibit transactivation of IRF7 of the IFN-β promoter suggests that bICP0 directly impacts IRF7 functions [48]. Interactions between bICP0 and p300 or p300-containing complexes may also interfere with IFN-β promoter activity because p300 is essential for stimulating IFN-β promoter activity [49]. BoHV-1 infection also impairs the phagocytic activity of macrophages and their antibody-dependent cellular cytotoxicity function [50,51].

BoHV-1, BoHV-5, and EHV-1 encode a glycoprotein G (gG) that is secreted from infected cells and binds to a broad range of chemokines [52]. Interactions between gG and chemokines block chemokine activity by preventing their interactions with specific receptors and glycosaminoglycans. By disrupting chemokine–glycosaminoglycan interactions, gG alters chemokine gradients, thereby influencing the local environment surrounding an infected cell. A BoHV-1 gG deletion mutant exhibits reduced virulence, in part because gG is a viral immune evasion gene [53]. In summary, BoHV-1 contains multiple means of impairing immune responses, which plays an important role in viral replication, pathogenesis, and transmission to other cattle.

## 4. Establishment and Maintenance of a Latent Infection in Sensory Neurons

### 4.1. Viral Gene Expression During Establishment and Maintenance of Latency

Viral particles produced during acute infection of mucosal surfaces enter the peripheral nervous system via cell–cell spread. Sensory neurons in trigeminal ganglia (TG) are a primary site for latency when infection is initiated in the oral, nasal, or ocular cavity [46,54,55,56,57]. Limited lytic cycle viral gene expression and virus production initially occur when sensory neurons are infected; however, sensory neurons do not support high levels of viral replication. Viral spread to the central nervous system occurs via synaptic connections from TG [58]. Viral gene expression is subsequently extinguished, and a subset of infected neurons survive. Infected neurons that survive initial infection harbor viral genomes, and latency is established [54]. Following the establishment of latency, viral shedding is not readily detected.

The latency-related (LR) gene is the only BoHV-1 gene abundantly expressed in latently infected neurons that overlaps with bICP0 coding sequences located in immediate early transcription unit 1 (IEtu1; Figure 1). The LR gene is complicated because polyA+ LR-RNA is alternatively spliced, and the start site for transcription is further upstream in the TG of latently infected calves relative to productive infection [59]. Since LR-RNA is alternatively spliced [60], it can encode several ORF2 family members, and ORF-1 if LR-RNA is not spliced. Part of reading frame B (RF-B) and RF-C may comprise the C-terminus of ORF-2 depending on how LR-mRNA is spliced. ORF-E [61] coding sequences are located in the LR promoter, and over-expression of ORF-E in murine neuroblastoma cells (Neuro-2A) promotes neurite sprouting and neuronal differentiation [62]. The LR gene locus also encodes two micro-RNAs located near the 5′-teminus of the LR-RNA transcript that are expressed during latency.

An LR mutant virus with three stop codons at the N-terminus of ORF2 exhibits reduced clinical symptoms in acutely infected calves because virus shedding from the TG, pharyngeal tonsil (PT), and ocular cavity of infected calves is reduced [63]. Wild-type (wt) BoHV-1, but not the LR mutant virus, reactivates from latency after DEX treatment [64], in part because the LR mutant induces high levels of TG neuronal death during the establishment of latency [65]. Since LR gene expression is repressed during reactivation from latency [17,66,67], they may not directly stimulate reactivation from latency. 

However, several LR gene products are predicted to promote the establishment and maintenance of latency. For instance, ORF2 interacts with certain cellular transcription factors: for example, Notch family members [68], a complex containing β-catenin and a β-catenin coactivator (High-Mobility Group AT-Hook 1 Protein) [69], and c/EBP-α [70]. ORF-2 interferes with Notch-1-mediated transcriptional activation [67,68,71,72], which may impair reactivation from latency because Notch is stimulated during this process [73]. ORF-2 promotes neuronal differentiation in Neuro-2A cells transfected with a Notch family member, which may promote the maintenance of latency because Notch family members impair neuronal differentiation to maintain a pool of neuronal stem cells [67]. ORF-2 also preferentially interacts with single-stranded DNA versus double-stranded DNA, but sequence-specific binding was not observed [74]. The anti-apoptosis activity of ORF2 is predicted to promote the survival of infected neurons, which is crucial for establishing and maintaining a life-long latent infection [75].

The LR gene and Herpes Simplex Virus 1 latency-associated transcript (LAT) both have anti-apoptosis functions [76,77]. Notably, a chimeric virus (CJ-LAT) that contains the BoHV-1 LR gene inserted at the location of the LAT in an LAT null mutant increased the incidence of reactivation from latency relative to the parental LAT null mutant (dLAT2903) [78,79]. These studies also revealed that CJLAT has enhanced virulence in mice and rabbits when compared to dLAT2903 and the parental wt McKRae strain. Expression of BoHV-1 ORF2 was crucial for these studies [78]. Over-expression of the LR gene micro-RNAs reduces steady-state levels of bICP0 proteins in transient transfection studies [17]. Another non-coding RNA in the LR gene promoter inhibits stress-induced activation of key BoHV-1 promoters [80]. In summary, LR gene products encode several factors that promote the establishment and maintenance of latency.

### 4.2. Activation of the Canonical β-Catenin/Wnt Signaling Axis Promotes the Establishment and Maintenance of Latency

RNA-sequencing studies demonstrated that the canonical Wnt/β-catenin signaling pathway is more active in the TG of latently infected calves when compared to TG from uninfected calves or during early stages of reactivation from latency [81]. The Wnt family is comprised of 19 known Wnt genes and 15 Wnt receptors or co-receptors [82,83]. Four distinct families of Wnt antagonists have also been identified. Interestingly, the levels of seven Wnt antagonists were significantly increased during the early stages of reactivation from latency by DEX [81].

If the Wnt pathway is active, Wnt family members interact with the Wnt co-receptor/receptor complex (Figure 2) [82,83]. These interactions increase disheveled (DVL) protein levels and the formation of a complex comprising glycogen synthase kinase 3β (GSK3β), casein kinase 1α (CK1α), Adenomatous polyposis coli gene (APC), and a scaffold protein Axin. Consequently, GSK3β is phosphorylated and its intrinsic kinase activity is turned off, culminating in increased cytoplasmic steady-state β-catenin protein levels. Unphosphorylated β-catenin in the cytosol subsequently enters the nucleus and interacts with T-cell-specific factor (TCF)/lymphoid enhancer-binding factor (LEF) and additional coactivators; consequently, β-catenin-dependent gene expression is induced. In the absence of a Wnt ligand, β-catenin interacts with the complex of GSK3β, Axin, APC, and CK1α. CK1α subsequently phosphorylates β-catenin, which leads to proteasome-dependent proteolysis and the inhibition of β-catenin-dependent gene expression.

Approximately 100 genes associated with the canonical Wnt/β-catenin signaling pathway are significantly reduced during DEX-induced reactivation from latency (30, 89) and 180 min after DEX treatment) [81]. The nucleotide-binding protein alpha-Q (GNAQ), (42-fold reduction), Wnt 16 (54-fold reduction), bone morphogenetic protein receptor 2 (BMPR2; 44-fold reduction), and Akt3 (51-fold reduction) were significantly reduced versus samples from latently infected calves. In general, these genes enhance canonical Wnt signaling via multiple mechanisms. Akt signaling stimulates the Wnt/β-catenin signaling pathway by phosphorylating β-catenin on serine 552, which enhances β-catenin-dependent transcription [84,85]. The Phosphoinositide 3-kinase signaling pathway (PI3K)/AKT signaling pathway also stimulates nuclear β-catenin localization [86]. Activated Akt interacts with the Axin–GSK3β complex in the presence of DVL, thus increasing β-catenin steady-state protein levels [87]. The three Akt family members are serine/threonine protein kinases that play significant roles in cell survival and growth, as reviewed in [88,89].

The expression of five soluble Wnt antagonists is significantly increased during reactivation from latency: for example, Dikkopf-1-like protein (DKKL1), Dikkopf-1 (DKK1), Wnt inhibitory factor 1 (WIF-1), and secreted frizzled-related protein 4 (SFRP4). Many Wnt regulators, positive and negative, are soluble secreted factors, suggesting they have multiple effects on TG neurons and support cells, regardless of whether they are latently infected.

The canonical Wnt/β-catenin signaling pathway (Figure 2) enhances neurogenesis and neuronal survival [90,91,92], crucial for a life-long latent infection following acute BoHV-1 infection. Based on these observations, we predict that the ability of BoHV-1 LR gene products to directly or indirectly co-opt the PI3K/Akt/Wnt/β-catenin signaling axis is crucial for mediating the establishment and maintenance of latency. Notably, several studies have concluded that the PI3K signaling pathway is linked to the Herpes Simplex Virus 1 latency–reactivation cycle [93,94].

## 5. Stress-Induced Reactivation from Latency Leads to Rapid Changes in Cellular Signaling Pathways That Correlate with Viral Gene Expression

### 5.1. Sensory Neurons in Trigeminal Ganglia Are Important Sites for BoHV-1 Latency and Reactivation from Latency

Stress, as mimicked by the synthetic corticosteroid hormone DEX, consistently induces BoHV-1 reactivation from latency [57,66,95]. Notably, synthetic corticosteroids also stimulate reactivation from latency in beagles latently infected with canine herpesvirus 1 [96,97,98,99]. Furthermore, DEX stimulates Herpes Simplex Virus 1 reactivation in TG explants [100,101]. Generally, a stressful stimulus triggers cortisol secretion via the hypothalamic–pituitary–adrenocortical (HPA) axis, as reviewed in [102]. Corticosteroids regulate cell growth, development, metabolism, and in certain situations, apoptosis. Inactive GR is engaged with a heat shock protein (HSP) complex located in the cytoplasm, as reviewed in [102,103] (Figure 3A). Increased cortisol levels lead to enhanced diffusion into cells. The GR–hormone complex disengages from the HSP complex; consequently, the GR–hormone complex enters the nucleus. A GR–hormone homodimer binds to a consensus GR response element (GRE), remodels chromatin, and stimulates transcription via a ligand-dependent mechanism [104,105]. A GR monomer can also stimulate transcription by binding to certain ½ GREs [106,107]. GR activation occurs within minutes and does not require de novo protein synthesis, which is important for rapidly inducing BoHV-1 promoters. Notably, GR can also stimulate gene expression via a ligand-independent mechanism [108] (Figure 3B). GR must be phosphorylated at serine 134 [108] for this process to occur. Serine 134 is hyperphosphorylated following glucose starvation, oxidative stress, UV irradiation, and osmotic shock. GR can be phosphorylated by mitogen-associated protein kinases (MAPKs), cyclin-dependent kinases (CDKs), and glycogen synthase kinase 3 beta (GSK3β) [108]. It is likely that other protein kinases also phosphorylate GR. Corticosteroids exhibit anti-inflammatory and immune-suppressive effects, in part by inactivating AP-1 and NF-κB, which are two transcription factors that stimulate the expression of inflammatory cytokines, as reviewed in [109]. Finally, activated GR induces apoptosis in certain lymphocytes, which also contributes to immune suppression and the anti-inflammatory properties of GR.

### 5.2. Pharyngeal Tonsil Contains Viral DNA in Latently Infected Calves and Supports Virus Reactivation

Viruses or bacteria that infect the nasal, ocular, or oral cavity drain into the throat and have the potential to infect pharyngeal tonsil (PT) cells. The nasal–lacrimal duct system allows tears from ocular surfaces to drain into the nasal cavity and then the throat; hence, BoHV-1 shedding from ocular or nasal surfaces drain to the PT. BoHV-1 DNA is consistently detected in the PT of latently infected cattle [110,111,112]. The DNA of Pseudorabies virus [113] and canine herpesvirus 1 [114] are also detected in the PTs of their respective hosts. For these α-herpesvirinae subfamily members, their respective hosts were not shedding the infectious virus and thus fit the criteria for being latently or quiescently infected.

BoHV-1 lytic cycle viral gene expression is readily detected in the PT by in situ hybridization 6 h after latently infected calves are given an IV injection of DEX [112]. RNA-sequencing studies demonstrated that the essential BoHV-1 regulatory transcript (bICP4) is expressed within 30 min in the PT after DEX treatment of latently infected calves (Figure 3C) [110]. As expected, lytic cycle viral gene expression is not detected in the PT of calves latently infected with BoHV-1. Within 90 min after DEX treatment, bICP0 and bICP4 were the only viral genes detected (Figure 3C). Surprisingly, LR-RNA is not readily detected in the PT of latently infected calves but is abundantly detected in the TG of latently infected calves. By 3 h after DEX treatment, all viral genes were readily detected in the PT. Based on these studies, the PT fit the criteria for being a novel site for BoHV-1 latency and not a “smoldering persistent infection”. In addition to distinct differences in viral gene expression in TG versus PTs during early stages of reactivation from latency, additional differences were identified. RNA-sequencing of the total PT, but not TG, readily detects viral RNA during the early stages of reactivation from latency, suggesting more cells in the PT are latently infected.

During reactivation from latency, virus transmission via the PT may be more efficient than TG because virus shedding directly spreads to the oral cavity or lower respiratory tract. Reactivation from latency via TG neurons requires that viral particles travel anterogradely from TG toward nerve terminals where innervated cells are subsequently infected prior to virus shedding and transmission to other cattle.

### 5.3. Viral Gene Expression in Trigeminal Ganglia During Reactivation from Latency Differs from That in Pharyngeal Tonsils

The expression of key viral regulatory proteins (bICP0, bICP4, bICP22, and VP16), but not glycoprotein C or E, is detected in TG neurons within 90 min after DEX treatment [115,116,117]. Consistently, bICP0 and VP16 are detected prior to bICP4 and bICP22 (Figure 3C). The expression of several cellular transcription factors is stimulated during reactivation from latency [73]. These include promyelocytic leukemia zinc finger (PLZF), Slug, SPDEF (Sam-pointed domain containing Ets transcription factor), Krüppel-like transcription factor 4 (4), KLF15, KLF6, and GATA6. These stress-induced transcription factors stimulate productive infection and key viral promoters as discussed below [73,118,119,120,121,122,123]. Interestingly, certain KLF family members (KLF4, KLF6, KLF15, and PLZF) and Sp1 are detected in more TG neurons after latently infected calves are treated with DEX to initiate reactivation from latency [36,73].

### 5.4. Stress-Induced Transcription Factors Stimulate IEtu1 and bICP0 E Promoters

The IEtu1 promoter, which drives bICP0 and bICP4 protein expression [27,28,124] (Figure 4A), contains two consensus GR response elements (GREs) and is stimulated by DEX [117,121] (Figure 4B). GR stimulates KLF15 expression, these two transcription factors interact, and they cooperatively transactivate the IEtu1 promoter via a feed-forward transcription loop [120]. The BoHV-1 genome is GC-rich, and many viral promoters, including the IEtu1 promoter, contain Sp1 consensus binding sites and other GC-rich motifs, suggesting KLF binding sites span the entire viral genome [31]. Silencing expression of the KLF15 protein hindered Herpes Simplex Virus 1 productive infection, and KLF15 protein levels increase during productive infection. KLF15 was predominantly localized in the nucleus after BoHV-1 or Herpes Simplex Virus 1 infection of cultured cells [125]. When cells were transfected with a KLF15 promoter construct and subsequently infected with Herpes Simplex Virus 1, there was a significant increase in promoter activity. The Herpes Simplex Virus 1 ICP0 gene, and to a lesser extent, bICP0, transactivates the KLF15 promoter in the absence of other viral proteins [125]. Krüppel-like factors (KLFs) and specificity protein 1 (Sp1) are closely related zinc-finger proteins that bind GC- or CA-rich sequences and form a superfamily of transcription factors [31,126]. KLF proteins and Sp1 family members play a crucial role in the transcriptional machinery of eukaryotic cells and influence cell proliferation, apoptosis, differentiation, and neoplastic transformation, as reviewed in [31,126,127]. KLF4, like several other KLF family members, has pro-apoptotic and anti-apoptotic functions, which are dependent on the cell type and whether these cells are normal or cancerous [128].

The E2F family of transcription factors regulates the cell cycle. They contain a conserved DNA-binding domain, an acidic transcriptional activation domain, and a binding site for the retinoblastoma (Rb) protein [129]. The phosphorylation of Rb family members by cyclin-dependent kinase–cyclin complexes releases E2F family members. E2F1, E2F2, and E2F3 activate transcription, whereas other members repress transcription or have little effect on it [130,131,132]. Remarkably, consensus E2F binding sites are found in the promoters of many genes that regulate cell cycle progression [130]. Previous studies demonstrated that E2F family members stimulate replication and gene expression of *α*-herpesvirinae subfamily members. For instance, silencing E2F1 reduces BoHV-1 and Herpes Simplex Virus 1 replication in cultured cells, and E2F1 or E2F2 transactivate the IEtu1 and early bICP0 promoters [35,133,134]. Interestingly, E2F1 also activates the Herpes Simplex Virus 1 thymidine kinase promoter via a GC-rich motif, rather than a consensus E2F binding site [135]. Sp1 over-expression can also stimulate [136,137] apoptosis, and other studies demonstrated that Sp1 and Sp3 impair apoptosis [138]. Interestingly, apoptosis was reported to accelerate reactivation from latency [99,139] suggesting certain KLF family members and Sp1 and Sp3 expression can enhance reactivation by enhancing BoHV-1 [36] and Herpes Simplex Virus 1 replication [35] or inducing apoptosis.

Recent studies revealed that E2F2, unlike other E2F transcriptional activators (E2F1, E2F3a, and E2F3b), significantly enhanced the transactivation of the CRM-derived IEtu1 promoter that contains both glucocorticoid response elements (GREs) when synthetic corticosteroid DEX was added to mouse neuroblastoma (Neuro-2A) or African Green monkey kidney (CV-1) cell cultures. Novel tandem Sp1 binding sites within the IEtu1GREs CRM were identified (see Figure 4B). Mutating these tandem Sp1 binding sites within the IEtu1GREs CRM significantly reduced E2F2-mediated transactivation in Neuro-2A and established CV-1 [140]. Chromatin immunoprecipitation studies revealed that E2F2 occupies IEtu1 promoter sequences in cultured cells.

The bICP0 E promoter is also synergistically transactivated by GR and KLF4, both of which are pioneer transcription factors that bind silent chromatin and activate gene expression [119] (Figure 4C). KLF15 also transactivates the bICP0 E promoter, but it had no effect on the gC promoter [73]. GR can function as a “pioneer” transcription factor because it can interact with silent heterochromatin, and under certain circumstances, can remodel silent chromatin, culminating in increased transcription [141]. The ability of GR to function as a pioneer factor, it is typically associated with a cellular ATPase, BRG1, which associates with at least 10 other transcriptional coactivators [142]. KLF4 is also a pioneer transcription factor, which coregulates GR-mediated gene expression [143], and KLF4 expression is stimulated by heat stress [144]. Since it is likely that BoHV-1 DNA is organized as silent heterochromatin in latently infected cells, pioneer transcription factors are predicted to play a key role in initiating viral gene expression during early stages of reactivation. Corticosteroid-mediated immune repression [106,145,146] is also predicted to increase virus spread during reactivation from latency.

## 6. Correlation Between BoHV-1 Reactivation from Latency and Disease

BoHV-1 reactivation from latency does not generally cause serious recurrent disease. However, reactivation from latency transiently suppresses immune responses, disrupts the integrity of mucosal surfaces, and is essential for virus transmission. Thus, reactivation from latency can expedite Mannheimia haemolytica colonization and bacterial replication in the lower respiratory tract, leading to pneumonia. Current modified live vaccines used in the US readily reactivate from latency and trigger abortion [21,25] by the hematogenous spread of virus from the placenta to the liver via the umbilical vein, and then to all organs via fetal blood vessels [147]. For example, we performed a study with Pfizer (now Zoetis) where calves were vaccinated with a temperature-sensitive BoHV-1 modified live vaccine (RLB 106). When vaccination was performed in the nasal cavity, DEX treatment of latently infected calves consistently led to reactivation from latency, as judged by the shedding of the infectious virus from oral, nasal, and PT swabs [148]. The establishment of latency in TG or sacral dorsal root ganglia did not occur when calves were vaccinated intramuscularly. This study highlights complications associated with the reactivation of BoHV-1 from latency in current modified live vaccines in feedlots and dairy farms. Thus, mechanistic studies designed to identify cellular and viral factors that trigger the early stages of reactivation from latency may lead to new strategies for designing a modified live vaccine that does not reactivate from latency or that exhibits reduced incidence of reactivation from latency.

## 7. Conclusions

BoHV-1 continues to be a significant pathogen in the cattle industry because it is an important cofactor for BRD and it causes abortions. There are several commercially available modified live vaccines. However, the current modified live vaccines can reactivate BoHV-1 from latency and cause disease, including abortions. The ability of stress to induce BoHV-1 reactivation from latency in TG neurons and certain cells in pharyngeal tonsils continues to be a significant problem. Our findings show that GR directly stimulates viral gene expression and viral replication, directly linking stress to BoHV-1 reactivation from latency. Furthermore, our study revealed that pioneer transcription factors, including GR and KLF4, are crucial for activating viral gene expression during early stages of reactivation from latency. Since GR activation also interferes with normal antiviral responses, immune suppression enhances BoHV-1 virus shedding. Identifying viral and cellular genes that initiate reactivation will provide new therapeutic strategies designed to prevent reactivation from latency.

## Figures and Tables

**Figure 1 viruses-16-01675-f001:**
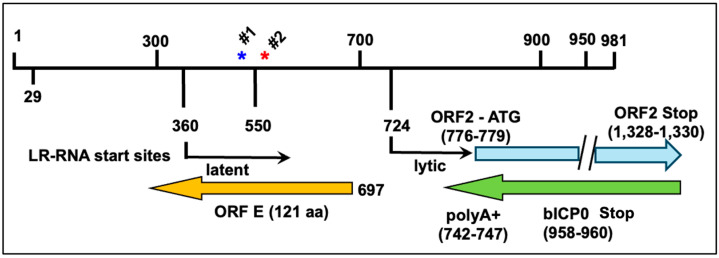
Schematic of latency-related (LR) gene products. Map of a Pst-I fragment containing the LR promoter, 5′ terminus of LR transcript, 3′-terminus of bICP0, and ORF-E gene. ORF-E is antisense relative to the LR gene. Arrow denotes start sites for LR-RNA transcription during latent and lytic infections (nucleotides 360 and 724). Position of LR micro-RNA #1 and #2 are denoted by stars. Stop codon for bICP0 translation and polyA+ addition site of bICP0 are shown. Location of ATG and ORF2 stop codon are also denoted. Nucleotide numbers denote the PstI 5′ terminus of the LR gene.

**Figure 2 viruses-16-01675-f002:**
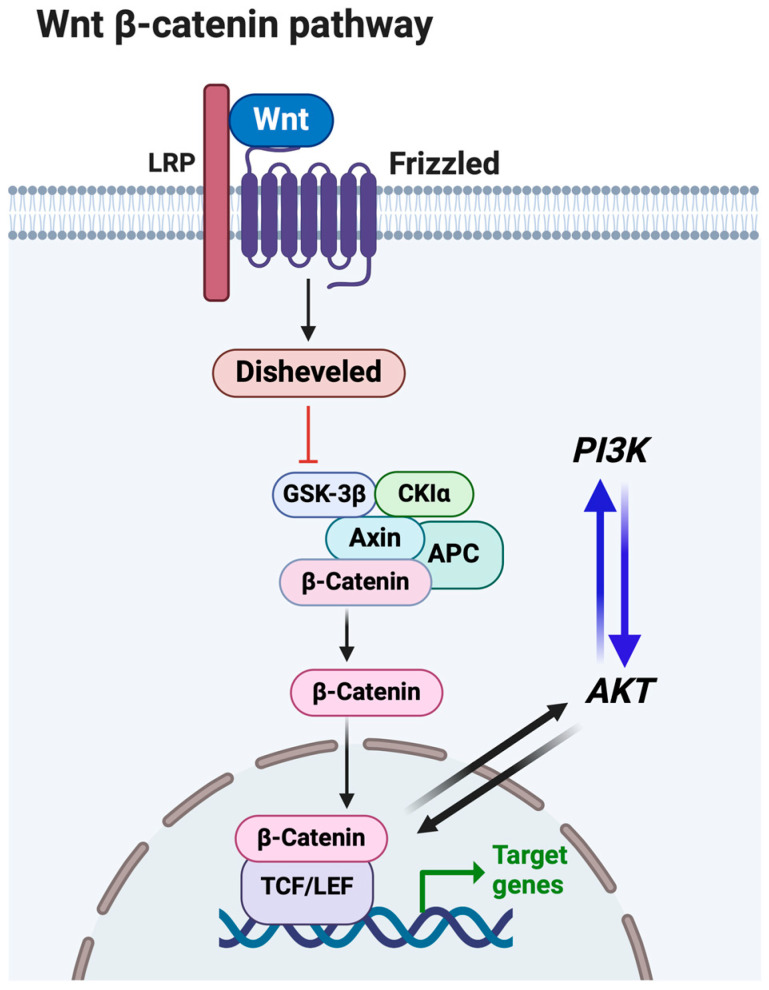
Schematic summarizing the canonical Wnt/β-catenin signaling pathway. For details, see the above, which describes this pathway.

**Figure 3 viruses-16-01675-f003:**
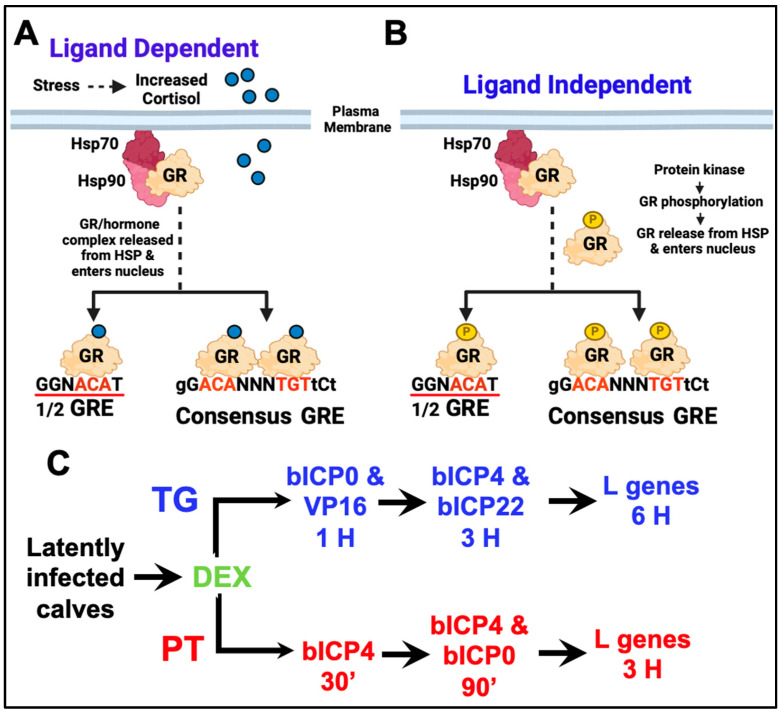
Stress-induced reactivation from latency. **Panel** (**A**): Schematic of key events that lead to GR activation by increased glucocorticoids secreted via the hypothalamic–pituitary axis (HPA). Red nucleotides in the GRE are essential nucleotides, capital letters are well-conserved nucleotides, small letters are flexible, and N can be any nucleotide. A GR–hormone dimer specifically binds to a consensus GRE. A GR–hormone homodimer can also bind to a ½ GRE (underlined in red) and stimulate transcription. **Panel** (**B**): Certain protein kinases described in the text can phosphorylate GR, which promotes the release of GR from the HSP complex (phosphorylated GR is denoted as GR-P). A phosphorylated GR dimer or GR monomer enters the nucleus, binds to a consensus GRE or ½ GRE, respectively, and transactivates promoters containing a ½ GRE. **Panel** (**C**): Comparison of viral gene expression in TG versus PT during early stages of DEX-induced reactivation from latency.

**Figure 4 viruses-16-01675-f004:**
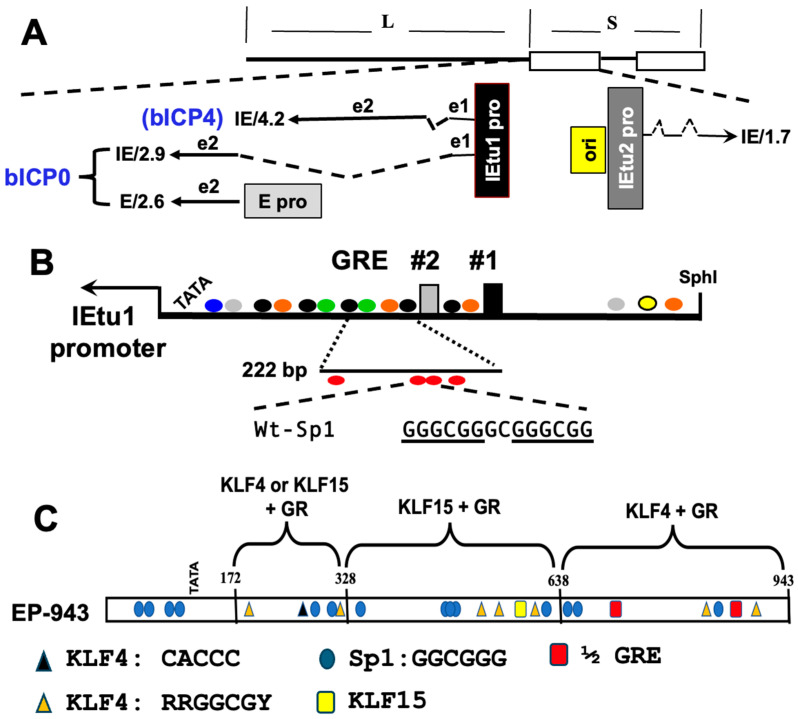
Schematic of IEtu1 promoter and bICP0 E promoter. **Panel** (**A**). The BoHV-1 genome contains a unique long (L), unique short (S), and two repeats denoted by the rectangles. Location of IE transcripts and LR transcript (LR-RNA) are denoted. IE/4.2 is the IE bICP4 transcript. IE/2.9 is the IE mRNA that encodes the bICP0 transcript. The IEtu1 promoter activates expression of IE/4.2 and IE/2.9 and is denoted by a black rectangle (IEtu1 pro). E/2.6 is the early transcript that encodes bICP0, and a separate early promoter regulates the expression of the E/2.6 transcript (E pro). Exon 2 (e2) of bICP0 contains all the protein-coding sequences. The origin of replication (ORI) is located between IEtu1 and IEtu2. The IEtu2 promoter (IEtu2 pro) controls the expression of the bICP22 protein. Solid lines in the transcript are exons (e1, e2, e3) and dashed lines are introns. **Panel** (**B**): Schematic of full-length IEtu1 promoter. Location of GREs, TATA box, and the start site for transcription are denoted by the arrow, and potential binding sites are shown for cellular transcription factors. **Panel** (**C**): Schematic of bICP0 E promoter and location of potential cellular transcription binding sites. Blue ovals denote consensus Sp1 binding sites, black triangles are consensus KLF CACCC-rich motif, orange triangles are KLF4-like binding sites, and red ovals are ½ GRE-like binding sites. The yellow oval is a potential KLF15 binding site.

## Data Availability

Not applicable.

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
