# Peer review of "Stress Can Induce Bovine Alpha-Herpesvirus 1 (BoHV-1) Reactivation from Latency"

_viruses, 2024, doi:10.3390/v16111675_

Round 1

Reviewer 1 Report

Comments and Suggestions for Authors

This review provides a nice summary of latency in bovine herpes virus infection. The authors provide a detailed and comprehensive review of BoHV-1 latency-reactivation. Overall, the manuscript provides new and important information, and the information are highly significant with regards to control of BoHV-1 infection and pathogenesis. This information can be used to study many new aspects of BoHV-1 infection-replication including the potential of using their approach to block viral replication and reactivation. Although this review is very interesting and important, there are several points that need to be addressed before publishing this review as follow:

1.    Title is too long. Can the authors shorting their title?

2.    Can the authors discuss their finding in relation to other members of Herpesviridae?

3.    The authors talk about the role of KLF family in BoHV-1 infection.  KLFs are belong to SP1 family and they are involved in apoptosis. Since LAT has anti-apoptotic function, can the authors speculate on the roles of KLF in apoptosis.

4.    Previously the authors shown differences between male and female mice with regards to SP1 expression. Did the authors observed similar results with calf? Does KLF play any role in these differences?

5.    Previously, the authors constructed CJ-LAT. Please discuss this important recombinant virus in detail in the revised manuscript.

Author Response

With respect to Reviewer #1, the following changes were made.

Concern #1: Title is too long. Can the authors shorting their title?

Response:  The title was shortened from 19 words to 10.  

Concern #2:  Can the authors discuss their finding in relation to other members of Herpesviridae?

Response:  This is discussed on lines 263-269.

Concern #3.    The authors talk about the role of KLF family in BoHV-1 infection.  KLFs are belong to SP1 family and they are involved in apoptosis. Since LAT has anti-apoptotic function, can the authors speculate on the roles of KLF in apoptosis.

Response:  These interesting points are discussed in the revised manuscript (line 604-606 and 367-372 and 386-391).

Concern #4.    Previously the authors shown differences between male and female mice with regards to SP1 expression. Did the authors observed similar results with calf? Does KLF play any role in these differences?

Response:  Interesting question, but these studies were not performed.  We primarily use Holstein calves for our calf studies.  Our sources will sell males but not females because females are kept for producing milk.  In general, it is very hard to purchase female calves for our bovine studies. 

Concern #5.    Previously, the authors constructed CJ-LAT. Please discuss this important recombinant virus in detail in the revised manuscript.

Response:  This topic is discussed in the revised manuscript (lines 176-187).

Reviewer 2 Report

Comments and Suggestions for Authors

The review summarizes the current knowledge on stress-induced transcription factors in BoHV-1 reactivation from latency. It will be a helpful starting point for researchers working on BoHV-1 latency and reactivation from latency. However, there is extensive overlap with another recent review by the group of Prof. Clinton Jones (PMID: 36851767) and even some figures (Fig. 1 and Fig. 2) and paragraphs overlap in these two reviews. Therefore the added value of this review to the field is not clear to me. The authors should refocus their review and not merely repeat what has been published before. Please find my additional comments below.

Major comments:

The review only briefly touches upon the viral pathogenesis, so the title does not fully represent the review. Either remove the word ‘pathogenesis’ from the title or discuss the viral pathogenesis in more detail so that the title better represents what is discussed in the article.

As a reader it was sometimes challenging to follow the article, especially since so many abbreviations are used, even in subtitles. I propose to remove any abbreviations from the subtitles and to minimize the number of abbreviations used in the text.

Lines 38-56: Please change this subtitle to general pathogenesis in cattle because also abortion is mentioned in this paragraph which has nothing to do with BRD. Please also include a more updated reference since many other pathogens are involved with BRD besides the ones mentioned here. It seems now that only Mannheimia heamolytica is involved in BRD as bacterial pathogens, while there are many others involved as well. It would be great if some more information on the pathogenesis is mentioned here, especially on where BoHV-1 establishes primary infection, secondary infection, viremia and latency.

Figure 1: This figure has already been used in PMID: 36851767 so please refer to this paper instead of republishing.

Figure 2: This figure has also already been used in the same review. Also the legend is currently missing.

Paragraph 6: the title does not fully represent the paragraph, please rephrase.

The review would benefit from a schematic showing the interaction between cellular and viral proteins and transcription factors involved in reactivation from latency by BoHV-1.

Minor comments

Line 67: HSV-1 should be spelled out as herpes simplex virus 1 here and not only in line 72.

Line 104-105: A less virulent virus does not necessarily have less immune evasion strategies. Please rephrase.

Line 250: typo “binding”

Line 384: clarify what MLV stands for

Author Response

With respect to Reviewer #2. the following changes were made

Concern #1: The review only briefly touches upon the viral pathogenesis, so the title does not fully represent the review. Either remove the word ‘pathogenesis’ from the title or discuss the viral pathogenesis in more detail so that the title better represents what is discussed in the article.

Response: The word pathogenesis was removed from the title.  Since this review is focused on latency and reactivation from latency, we briefly summarized the pathogenesis section.

Concern #2: As a reader it was sometimes challenging to follow the article, especially since so many abbreviations are used, even in subtitles. I propose to remove any abbreviations from the subtitles and to minimize the number of abbreviations used in the text.

Response:  The abbreviations were used to shorten the manuscript.  I believe we used minimal abbreviations in this review.  I have spelled out HSV-1 and modified live vaccines throughout in the revised manuscript.  However, I do not feel we should spell out every scientific term; for example, bICP0, IEtu1, etc throughout the manuscript.  The first time these abbreviations were used the term is spelled out.

Concern #3:  Lines 38-56: Please change this subtitle to general pathogenesis in cattle because also abortion is mentioned in this paragraph which has nothing to do with BRD. Please also include a more updated reference since many other pathogens are involved with BRD besides the ones mentioned here.

Response:  Section 1 is now referred to as Pathogenesis of BoHV-1The main infectious agents that are known to be cofactors in BRD are mentioned.  Hence, I do not feel there is a need to add additional references. 

Concern #4: It seems now that only Mannheimia heamolytica is involved in BRD as bacterial pathogens, while there are many others involved as well. It would be great if some more information on the pathogenesis is mentioned here, especially on where BoHV-1 establishes primary infection, secondary infection, viremia and latency.

            Response:  The most common bacterial strain associated with BRD are now listed in the revised manuscript (lines 44-48).  Additional information about when BoHV-1 plays a role in BRD is also mentioned (lines 52-57). 

Concern #5: Figure 1: This figure has already been used in PMID: 36851767 so please refer to this paper instead of republishing.

            Response:  I did not realize this.  A new Figure 1 that only focuses on the BoHV-1 LR gene is now presented.  .

Concern #6: Figure 2: This figure has also already been used in the same review. Also the legend is currently missing.

            Response:  This is not an accurate statement.  I pasted a screen shot of this figure from the 2023 review (PMID: 36851767).  The Figure 2 of this review is not the figure in the 2023 review.

Concern #7: Paragraph 6: the title does not fully represent the paragraph, please rephrase.

The review would benefit from a schematic showing the interaction between cellular and viral proteins and transcription factors involved in reactivation from latency by BoHV-1.

            Response:  I am assuming this means Section 6This section is now called: Correlation between BoHV-1 reactivation from latency and disease (line 419).  This section was added because most people do not believe BoHV-1 can cause recurrent disease.  However, the ability of BoHV-1 to reactivate from latency (field strains or modified live vaccines) can lead to health issues (lines 420-439). 

Concern #8: Line 67: HSV-1 should be spelled out as herpes simplex virus 1 here and not only in line 72.

            Response:  I spelled out Herpes Simplex Virus 1 throughout the text but not in the references. 

Concern #9: Line 104-105: A less virulent virus does not necessarily have less immune evasion strategies. Please rephrase.

            Response:  This sentence was referenced by the following manuscript: Bryant NA, N. Davis-Poynter, A. Vanderplasschen, and A. Alcami 2003. Glycoprotein G isoforms from some alphaherpesvirus function as broad-spectrum chemokine binding proteins. EMBO J 22:833-846.

Since Glycoprotein G is a broad-spectrum chemokine binding proteins, it is reasonable to suggest gG has antiviral functions.  It is well established that chemokines are signaling proteins secreted by cells that attract naïve and activated lymphocytes.  These important points are made in the revised manuscript (lines 107-116). 

Concern #10: Line 250: typo “binding”

Response:  This was fixed (lines 277). 

Concern #11: Line 384: clarify what MLV stands for

Response:  All 4 MLVs were changed to modified live vaccines (lines 425-438 and 443). 

Round 2

Reviewer 2 Report

Comments and Suggestions for Authors

The review has been adapted well. I only have a few minor issues:

General comment: The current review is quite similar to a previous review (PMID 36851767). However, this review is not mentioned in the current review. Therefore, please point out how the current review is new to the field and include it as a reference.

Concern #2: As a reader it was sometimes challenging to follow the article, especially since so many abbreviations are used, even in subtitles. I propose to remove any abbreviations from the subtitles and to minimize the number of abbreviations used in the text.

Response:  The abbreviations were used to shorten the manuscript.  I believe we used minimal abbreviations in this review.  I have spelled out HSV-1 and modified live vaccines throughout in the revised manuscript.  However, I do not feel we should spell out every scientific term; for example, bICP0, IEtu1, etc throughout the manuscript.  The first time these abbreviations were used the term is spelled out.

Response reviewer: In subtitle ‘Viral gene expression in trigeminal ganglia during reactivation from latency differs from PT’ I suggest to also write PT in full, like you do for trigeminal ganglion.

Also in Fig. 1 write LR in full (not mentioned above yet).

Concern #3:  Lines 38-56: Please change this subtitle to general pathogenesis in cattle because also abortion is mentioned in this paragraph which has nothing to do with BRD. Please also include a more updated reference since many other pathogens are involved with BRD besides the ones mentioned here.

Response:  Section 1 is now referred to as Pathogenesis of BoHV-1The main infectious agents that are known to be cofactors in BRD are mentioned.  Hence, I do not feel there is a need to add additional references. 

Response reviewer: Reference 1 dates from 2007. Since then, other viral pathogens have also been identified to play a role in BRD (e.g., bovine coronaviruses, influenza D). Even parainfluenzavirus type 3 is not mentioned. Please update this sentence and reference.

Typo line 33: common bacterial strain à strains

Line 46-48: Please provide a reference to confirm this statement.

Typo line 107: genes … establishes and maintains à establish and maintain

Author Response

With respect to Reviewer #2, the following changes have been made in the revised manuscript.

Concern #1: General comment: The current review is quite similar to a previous review (PMID 36851767). However, this review is not mentioned in the current review. Therefore, please point out how the current review is new to the field and include it as a reference.

                  Response: This information is included; see lines 38-47.

Concern #2: In subtitle ‘Viral gene expression in trigeminal ganglia during reactivation from latency differs from PT’ I suggest to also write PT in full, like you do for trigeminal ganglion.

                  Response: As suggested, PT is now referred to as pharyngeal tonsil (line 338).

Concern #3: Also in Fig. 1 write LR in full (not mentioned above yet).

Response: As suggested, LR is now referred to Latency Related (line 131).

Concern #4: Reference 1 dates from 2007. Since then, other viral pathogens have also been identified to play a role in BRD (e.g., bovine coronaviruses, influenza D). Even parainfluenzavirus type 3 is not mentioned. Please update this sentence and reference.

Response: bovine coronaviruses and influenza D are now included in the revised manuscript and there is a 2022 reference; see lines 49-52

Concern #5: Typo line 33: common bacterial strain à strains

Response: strains is now stated instead of strain (line 54-56).

Concern #6: Line 46-48: Please provide a reference to confirm this statement. 

Response:  This was done; line 56.

Concern #6: Typo line 107: genes … establishes and maintains à establish and maintain

Response:  this typographical error was fixed (line 141).